# Fast Minimum-norm Adversarial Attacks through Adaptive Norm Constraints

**Maura Pintor** [1 2]  **Fabio Roli** [1 2]  **Wieland Brendel** [3]  **Battista Biggio** [1 2]

## Abstract

Evaluating adversarial robustness amounts to finding the minimum perturbation needed to have an input sample misclassified. The inherent complexity of the underlying optimization requires current gradient-based attacks to be carefully tuned, initialized, and possibly executed for many computationally-demanding iterations, even if specialized to a given perturbation model. In this work, we overcome these limitations by proposing a fast minimum-norm (FMN) attack that works with different $\ell_p$-norm perturbation models ($p = 0, 1, 2, \infty$), is robust to hyperparameter choices, does not require adversarial starting points, and converges within few lightweight steps. It works by iteratively finding the sample misclassified with maximum confidence within an $\ell_p$-norm constraint of size $\epsilon$, while adapting $\epsilon$ to minimize the distance of the current sample to the decision boundary. Extensive experiments show that FMN significantly outperforms existing attacks in terms of convergence speed and computation time, while reporting comparable or even smaller perturbation sizes.

## 1. Introduction

Learning algorithms are vulnerable to adversarial examples, i.e., intentionally-perturbed inputs aimed to mislead classification at test time (Szegedy et al., 2014; Biggio et al., 2013). While adversarial examples have received much attention, evaluating the robustness of deep networks against them remains a challenge. Having an arsenal of diverse attacks that can be adapted to specific defenses is one of the most promising avenues for increasing confidence in white-box robustness evaluations (Carlini et al., 2019; Tramer et al., 2020). While it may seem that the number of attacks is

already large, most of them are just small variations of the same technique, make similar underlying assumptions and thus tend to fail jointly.

In this work, we focus on *minimum-norm* attacks for evaluating adversarial robustness, i.e., attacks that aim to mislead classification by finding the smallest input perturbation according to a given norm. Within the class of gradient-based minimum-norm attacks, there are three main subcategories: (i) soft-constraint attacks, (ii) boundary attacks and (iii) projected-gradient attacks. Soft-constraint attacks like CW (Carlini & Wagner, 2017) optimize a trade-off between confidence of the misclassified samples and perturbation size. CW needs a sample-wise tuning of the trade-off hyperparameter to find the smallest possible perturbation, thus requiring many steps to converge. Boundary attacks like BB (Brendel et al., 2019) or FAB (Croce & Hein, 2020b) move along the decision boundary towards the closest point to the input sample. These attacks converge within relatively few steps, but may require an adversarial starting point and need to solve expensive optimization problems in each step. Finally, recent minimum-norm projected-gradient attacks like DDN (Rony et al., 2019) perform a maximum-confidence attack in each step under a given perturbation budget $\epsilon$, while iteratively adjusting $\epsilon$ to reduce the perturbation size, however, DDN is specific to the $\ell_2$ norm and cannot be readily extended to other norms.

To overcome the aforementioned limitations, in this work we propose a novel, fast minimum-norm (FMN) attack (Sect. 2), which retains the main advantages of DDN while generalizing it to different $\ell_p$ norms ($p = 0, 1, 2, \infty$). We perform large-scale experiments (Sect. 3), showing that FMN is able to combine all desirable traits a good adversarial attack should have, providing an important step towards improving adversarial robustness evaluations. We conclude by discussing limitations and future research directions (Sect. 4).

## 2. Minimum-Norm Adversarial Examples with Adaptive Projections

**Problem formulation.** Given an input sample $\boldsymbol{x} \in [0, 1]^d$, belonging to class $y \in \{1, \dots, c\}$, the goal of an untargeted attack is to find the minimum-norm perturbation $\boldsymbol{\delta}^\star$ such that the corresponding adversarial example $\boldsymbol{x}^\star = \boldsymbol{x} + \boldsymbol{\delta}^\star$ is

---
*Equal contribution [1]Department of Electrical and Electronic Engineering, University of Cagliari, Italy [2]Pluribus One [3]University of Tübingen, Germany. Correspondence to: Maura Pintor <maura.pintor@unica.it>.

*Accepted by the ICML 2021 workshop on A Blessing in Disguise: The Prospects and Perils of Adversarial Machine Learning.* Copyright 2021 by the author(s).

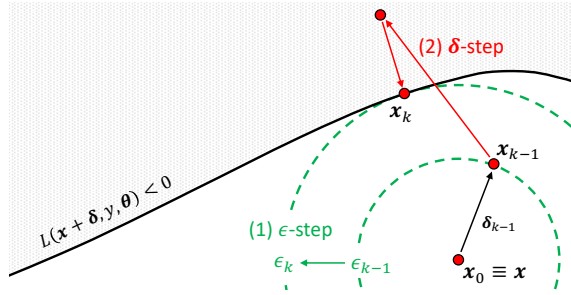

*Figure 1.* Conceptual representation of the FMN attack algorithm. The $\epsilon$-step updates the constraint size $\epsilon$ to minimize its distance to the boundary. The $\boldsymbol{\delta}$-step updates the perturbation $\boldsymbol{\delta}$ with a projected-gradient step to maximize misclassification confidence within the current $\epsilon$-sized constraint.

misclassified. This problem can be formulated as:

$$\boldsymbol{\delta}^{\star} \in \arg\min_{\boldsymbol{\delta}} \quad \|\boldsymbol{\delta}\|_p \,, \tag{1}$$

$$\text{s.t.} \quad L(\boldsymbol{x} + \boldsymbol{\delta}, y, \boldsymbol{\theta}) < 0 \,, \tag{2}$$

$$\boldsymbol{x} + \boldsymbol{\delta} \in [0, 1]^d \,, \tag{3}$$

where $\|\cdot\|_p$ indicates the $\ell_p$-norm operator. The loss $L$ in the constraint in Eq. (2) is defined as:

$$L(\boldsymbol{x}, y, \boldsymbol{\theta}) = f_y(\boldsymbol{x}, \boldsymbol{\theta}) - \max_{j \neq y} f_j(\boldsymbol{x}, \boldsymbol{\theta}) \,, \tag{4}$$

where $f_j(\boldsymbol{x}, \boldsymbol{\theta})$ is the confidence given by the model $f$ for classifying $\boldsymbol{x}$ as class $j$, and $\boldsymbol{\theta}$ is the set of its learned parameters. Assuming that the classifier assigns $\boldsymbol{x}$ to the class exhibiting the highest confidence, i.e., $y^{\star} = \arg\max_{j \in 1,\ldots,c} f_j(\boldsymbol{x}, \boldsymbol{\theta})$, the loss function $L(\boldsymbol{x}, y, \boldsymbol{\theta})$ takes on negative values only when $\boldsymbol{x}$ is misclassified[1]. Finally, the box constraint in Eq. (3) ensures that the perturbed sample $\boldsymbol{x} + \boldsymbol{\delta}$ lies in the feasible input space.

**Solution algorithm.** To solve Problem (1)-(3), we reformulate it using an upper bound $\epsilon$ on $\|\boldsymbol{\delta}\|_p$:

$$\min_{\epsilon, \boldsymbol{\delta}} \epsilon \,, \quad \text{s.t.} \|\boldsymbol{\delta}\|_p \leq \epsilon \,, \tag{5}$$

and to the constraints in Eqs. (2)-(3). This allows us to derive an algorithm that works in two main steps, similarly to DDN (Rony et al., 2019), by updating the maximum perturbation size $\epsilon$ separately from the actual perturbation $\boldsymbol{\delta}$, as represented in Fig. 1. In particular, the constraint size $\epsilon$ is adapted to reduce the distance of the constraint to the boundary ($\epsilon$-step), while the perturbation $\boldsymbol{\delta}$ is updated using a projected-gradient step to minimize the loss function $L$ within the given $\epsilon$-sized constraint ($\boldsymbol{\delta}$-step). The complete algorithm is given as Algorithm 1, while a more detailed explanation of the two aforementioned steps is given below.

---

[1]This can be extended to the targeted case by modifying the loss function in Eq. (4) as $L^t(\boldsymbol{x}, y', \boldsymbol{\theta}) = \max_{j \neq y'} f_j(\boldsymbol{x}, \boldsymbol{\theta}) - f_{y'}(\boldsymbol{x}, \boldsymbol{\theta}) = -L(\boldsymbol{x}, y', \boldsymbol{\theta})$, i.e., changing its sign and using the target class label $y'$ instead of the true class label $y$

---

**Algorithm 1** Fast Minimum-norm (FMN) Attack

**Input:** $\boldsymbol{x}$, the input sample; $t$, a variable denoting whether the attack is targeted ($t = +1$) or untargeted ($t = -1$); $y$, the target (true) class label if the attack is targeted (untargeted); $\gamma_0$ and $\gamma_K$, the initial and final $\epsilon$-step sizes; $\alpha_0$ and $\alpha_K$, the initial and final $\boldsymbol{\delta}$-step sizes; $K$, the number of iterations.

**Output:** The minimum-norm adversarial example $\boldsymbol{x}^{\star}$.

1: $\boldsymbol{x}_0 \leftarrow \boldsymbol{x}$, $\epsilon_0 = 0$, $\boldsymbol{\delta}_0 \leftarrow \boldsymbol{0}$, $\boldsymbol{\delta}^{\star} \leftarrow \infty$
2: **for** $k = 1, \ldots, K$ **do**
3: $\quad \boldsymbol{g} \leftarrow t \cdot \nabla_{\boldsymbol{\delta}} L(\boldsymbol{x}_{k-1} + \boldsymbol{\delta}, y, \boldsymbol{\theta})$ // *loss gradient*
4: $\quad \gamma_k \leftarrow h(\gamma_0, \gamma_K, k, K)$ // *$\epsilon$-step size decay*
5: $\quad$ **if** $L(\boldsymbol{x}_{k-1}, y, \boldsymbol{\theta}) \geq 0$ **then**
6: $\quad\quad$ **if** *adversarial not found yet* **then**
7: $\quad\quad\quad \epsilon_k = \|\boldsymbol{\delta}_{k-1}\|_p + L(\boldsymbol{x}_{k-1}, y, \boldsymbol{\theta})/\|\boldsymbol{g}\|_q$
8: $\quad\quad$ **else**
9: $\quad\quad\quad \epsilon_k = \epsilon_{k-1}(1 + \gamma_k)$
10: $\quad\quad$ **end if**
11: $\quad$ **else**
12: $\quad\quad$ **if** $\|\boldsymbol{\delta}_{k-1}\|_p \leq \|\boldsymbol{\delta}^{\star}\|_p$ **then**
13: $\quad\quad\quad \boldsymbol{\delta}^{\star} \leftarrow \boldsymbol{\delta}_{k-1}$ // *update best min-norm solution*
14: $\quad\quad$ **end if**
15: $\quad\quad \epsilon_k = \min(\epsilon_{k-1}(1 - \gamma_k), \|\boldsymbol{\delta}^{\star}\|_p)$
16: $\quad$ **end if**
17: $\quad \alpha_k \leftarrow h(\alpha_0, \alpha_K, k, K)$ // *$\boldsymbol{\delta}$-step size decay*
18: $\quad \boldsymbol{\delta}_k \leftarrow \boldsymbol{\delta}_{k-1} + \alpha_k \cdot \boldsymbol{g}/\|\boldsymbol{g}\|_2$
19: $\quad \boldsymbol{\delta}_k \leftarrow \Pi_{\epsilon}(\boldsymbol{x}_0 + \boldsymbol{\delta}_k) - \boldsymbol{x}_0$
20: $\quad \boldsymbol{\delta}_k \leftarrow \text{clip}(\boldsymbol{x}_0 + \boldsymbol{\delta}_k) - \boldsymbol{x}_0$
21: $\quad \boldsymbol{x}_k \leftarrow \boldsymbol{x}_0 + \boldsymbol{\delta}_k$
22: **end for**
23: **return** $\boldsymbol{x}^{\star} \leftarrow \boldsymbol{x}_0 + \boldsymbol{\delta}^{\star}$

---

$\epsilon$**-step.** This step updates the upper bound $\epsilon$ on the perturbation norm (lines 4-16 in Algorithm 1). The underlying idea is to increase $\epsilon$ if the current sample is not adversarial (i.e., $L(\boldsymbol{x}_{k-1}, y, \boldsymbol{\theta}) \geq 0$), and to decrease it otherwise, while reducing the step size to dampen oscillations around the boundary and reach convergence. In the former case ($\epsilon$-*increase*), the increment of $\epsilon$ depends on whether an adversarial example has been previously found or not. If not, we estimate the distance to the boundary with a first-order linear approximation, and set $\epsilon_k = \|\boldsymbol{\delta}_{k-1}\|_p + L(\boldsymbol{x}_{k-1}, y, \boldsymbol{\theta})/\|\nabla L(\boldsymbol{x}_{k-1}, y, \boldsymbol{\theta})\|_q$, being $q$ the dual norm of $p$. This approximation allows the attack point to make faster progress towards the decision boundary. Conversely, if an adversarial sample has been previously found, but the current sample is not adversarial, it is likely that the current estimate of $\epsilon$ is only slightly smaller than the minimum-norm solution. We thus increase $\epsilon$ by a small fraction as $\epsilon_k = \epsilon_{k-1}(1 + \gamma_k)$, being $\gamma_k$ a decaying step size. In the latter case ($\epsilon$-*decrease*), if the current sample is adversarial, i.e., $L(\boldsymbol{x}_{k-1}, y, \boldsymbol{\theta}) < 0$, we decrease $\epsilon$ as

$\epsilon_k = \epsilon_{k-1} (1 - \gamma_k)$, to check whether the current solution can be improved. If the corresponding $\epsilon_k$ value is larger than the optimal $\|\boldsymbol{\delta}^\star\|_p$ found so far, we retain the best value and set $\epsilon_k = \|\boldsymbol{\delta}^\star\|_p$. To ensure convergence, the step size $\gamma_k$ is decayed with cosine annealing.

**$\boldsymbol{\delta}$-step.** This step updates $\boldsymbol{\delta}$ (lines 17-21 in Algorithm 1). The goal is to find the adversarial example that is misclassified with maximum confidence (i.e., for which $L$ is minimized) within the current $\epsilon$-sized constraint (Eq. 5) and bounds (Eq. 3). This amounts to performing a projected-gradient step along the negative gradient of $L$. We consider a normalized steepest descent with decaying step size $\alpha$ to overcome potential issues related to noisy gradients while ensuring convergence (line 18). The step size $\alpha$ is decayed using cosine annealing. Once $\boldsymbol{\delta}$ is updated, we project it onto the given $\epsilon$-sized $\ell_p$-norm constraint via a projection operator $\Pi_\epsilon$ (line 19), to fulfill the constraint in Eq. (5). The projection is trivial for $p = \infty$ and $p = 2$. For $p = 1$, we use the efficient algorithm by Duchi et al. (2008). For $p = 0$, we retain only the first $\epsilon$ components of $\boldsymbol{\delta}$ exhibiting the largest absolute value. We finally clip the components of $\boldsymbol{\delta}$ that violate the bounds in Eq. (3) (line 20).

**Adversarial initialization.** Our attack can be initialized from the input sample $\boldsymbol{x}$, or from a point $\boldsymbol{x}_{\text{init}}$ belonging either to a different class (untargeted attacks) or to the target class (targeted attacks). When initializing the attack from $\boldsymbol{x}_{\text{init}}$, we perform a 10-step binary search between $\boldsymbol{x}$ and $\boldsymbol{x}_{\text{init}}$, to find an adversarial point closer to the boundary.

**Differences with DDN.** FMN applies substantial changes to both the algorithm and the formulation of DDN. The main difference is that (i) DDN always rescales the perturbation to have size $\epsilon$. This operation is problematic when using other norms, especially sparse ones, as it hinders the ability of the attack to explore the neighboring space and find a suitable descent direction; (ii) FMN uses the logit difference as the loss function $L$; (iii) FMN does not need an initial value for $\epsilon$, as $\epsilon$ is dynamically estimated; and (iv) $\gamma$ is decayed to improve convergence around better minimum-norm solutions, by more effectively dampening oscillations around the boundary. Finally, we include the possibility of (v) initializing the attack from an adversarial point, which can greatly increase the convergence speed.

## 3. Experiments

We report here an extensive experimental analysis involving several state-of-the-art defenses and minimum-norm attacks, covering $\ell_0$, $\ell_1$, $\ell_2$ and $\ell_\infty$ norms. The goal is to empirically benchmark our attack and assess its effectiveness and efficiency as a tool for adversarial robustness evaluation.

### 3.1. Experimental Setup

**Datasets and models.** We consider two commonly-used datasets for benchmarking adversarial robustness of deep neural networks, i.e., the MNIST handwritten digits and CIFAR10. Following the experimental setup in Brendel et al. (2019), we use a subset of 1000 test samples to evaluate the considered attacks and defenses. We use a diverse selection of models to thoroughly evaluate attacks under different conditions. For MNIST, we consider the following four models: *M1*, a 9-layer undefended ConvNet; *M2*, the robust model by Madry et al. (2017), trained on $\ell_\infty$ attacks; *M3*, the robust model by Rony et al. (2019), trained on $\ell_2$ attacks; and *M4*, the IBP Large Model by Zhang et al. (2020). For CIFAR10, we consider three state-of-the-art robust models from RobustBench (Croce et al., 2020): *C1*, the robust model by Madry et al. (2017), trained on $\ell_\infty$ attacks; *C2*, the defended model by Carmon et al. (2019), trained on $\ell_\infty$ attacks and additional unsupervised data; and *C3*, the robust model by Rony et al. (2019), trained on $\ell_2$ attacks.

**Attacks.** We compare our algorithm against different state-of-the-art attacks for finding minimum-norm adversarial perturbations across different norms: the Carlini & Wagner (CW) attack (Carlini & Wagner, 2017), the Decoupling Direction and Norm (DDN) attack (Rony et al., 2019), the Brendel & Bethge (BB) attack (Brendel et al., 2019), and the Fast Adaptive Boundary (FAB) attack (Croce & Hein, 2020b). All these attacks are defined on the $\ell_2$ norm. BB and FAB are also defined on the $\ell_1$ and $\ell_\infty$ norms, and only BB is defined on the $\ell_0$ norm. We consider untargeted and targeted attack scenarios, except for FAB, which is only evaluated in the untargeted case.[2]

To ensure a fair comparison, we perform an extensive hyperparameter search for each of the considered attacks. We consider *dataset-level* hyperparameter tuning, a scenario in which we choose the same hyperparameters for all samples, selecting the configuration that yields the best attack performance on a smaller validation set. The hyperparameter configurations considered for each attack are detailed in the appendix.

**Evaluation criteria.** We evaluate the attacks along two different criteria (additional experiments can be found in the Appendix): (i) *perturbation size* measured as the median $\|\boldsymbol{\delta}^\star\|_p$ on the test set; (ii) *execution time*, measured as the average time spent per query (in milliseconds).

---

[2]FAB implements a substantially different attack in the targeted case. The targeted version of FAB aims to find a closer untargeted misclassification by running the attack a number of times, each time targeting a different candidate class, and then selecting the best solution (Croce & Hein, 2020b;a).

*Table 1.* Median $\|\delta^\star\|_p$ value at $Q = 1000$ queries for targeted and untargeted attacks, with dataset-level hyperparameter tuning.

| | | MNIST | | | | | | | | CIFAR10 | | | | | |
| | | *Untargeted* | | | | *Targeted* | | | | *Untargeted* | | | *Targeted* | | |
| | Model | M1 | M2 | M3 | M4 | M1 | M2 | M3 | M4 | C1 | C2 | C3 | C1 | C2 | C3 |
|---|---|---|---|---|---|---|---|---|---|---|---|---|---|---|---|
| $\ell_0$ | BB | 12 | 152 | 52 | 145 | 20 | 179 | 39 | 183 | 28 | 44 | 32 | 29 | 65 | 33 |
| | Ours | **9** | **33** | **18** | **15** | **16** | **48** | **28** | **55** | **11** | **17** | **16** | **25** | **38** | **32** |
| $\ell_1$ | FAB | 8.66 | 225.7 | 163.9 | 312.3 | - | - | - | - | - | - | 20.48 | - | - | - |
| | BB | 10.60 | 49.83 | 17.57 | 46.99 | 16.60 | 53.11 | 29.89 | 54.31 | 7.02 | 10.20 | 17.13 | 11.41 | 15.26 | 23.37 |
| | Ours | **7.13** | **4.18** | **13.66** | **4.99** | **13.18** | **8.33** | **21.37** | **12.16** | **4.28** | **4.82** | **9.52** | **8.51** | **10.40** | **17.32** |
| $\ell_2$ | FAB | 1.54 | 1.59 | 2.81 | 16.30 | - | - | - | - | 0.77 | 1.11 | 1.06 | - | - | - |
| | CW | 1.63 | 5.15 | 3.71 | - | 2.50 | - | 4.72 | - | 0.86 | 1.00 | 0.99 | 1.36 | 2.90 | 1.55 |
| | BB | 1.75 | 1.82 | 3.02 | 4.57 | 2.64 | 2.59 | 3.52 | 5.31 | 0.86 | 0.95 | 1.10 | 1.25 | 1.45 | 1.73 |
| | DDN | **1.47** | 2.01 | 2.62 | **1.15** | 2.31 | 2.72 | 3.36 | **1.96** | **0.66** | 0.77 | 0.91 | 1.11 | 1.31 | 1.40 |
| | Ours | 1.61 | **1.42** | **2.61** | 1.56 | **2.30** | **2.13** | **3.24** | 2.41 | 0.67 | **0.74** | **0.91** | **1.09** | **1.28** | **1.38** |
| $\ell_\infty$ | FAB | .148 | .365 | .248 | .900 | - | - | - | - | .038 | .052 | .029 | - | - | - |
| | BB | .159 | **.336** | .243 | .409 | .223 | **.361** | .280 | .477 | .044 | .054 | .029 | .059 | .074 | .042 |
| | Ours | **.140** | .357 | **.233** | **.408** | **.206** | .426 | **.277** | **.434** | **.034** | **.042** | **.024** | **.057** | **.066** | **.037** |

*Table 2.* Average execution time (milliseconds / query) for each attack-model pair.

| | | MNIST | | | | | | | | CIFAR10 | | | | | |
| | | *Untargeted* | | | | *Targeted* | | | | *Untargeted* | | | *Targeted* | | |
| | Model | M1 | M2 | M3 | M4 | M1 | M2 | M3 | M4 | C1 | C2 | C3 | C1 | C2 | C3 |
|---|---|---|---|---|---|---|---|---|---|---|---|---|---|---|---|
| $\ell_0$ | BB | 10.76 | 11.85 | 10.19 | 12.02 | 60.88 | 62.17 | 62.31 | 57.74 | 46.51 | 50.31 | 50.43 | 99.71 | 105.28 | 103.53 |
| | Ours | **5.15** | **4.87** | **5.87** | **9.70** | **5.14** | **4.75** | **5.85** | **9.71** | **26.26** | **30.54** | **30.89** | **26.13** | **30.26** | **30.81** |
| $\ell_1$ | FAB | 9.38 | 8.88 | 12.61 | 36.00 | - | - | - | - | 84.04 | 108.91 | 108.64 | - | - | - |
| | BB | 6.73 | 7.03 | 7.31 | 12.50 | 43.25 | 43.54 | 43.69 | 43.86 | 32.56 | 37.40 | 37.59 | 68.99 | 73.33 | 74.03 |
| | Ours | **5.43** | **5.14** | **6.10** | **9.35** | **5.44** | **5.10** | **6.09** | **9.35** | **27.34** | **31.17** | **31.18** | **26.00** | **30.98** | **31.03** |
| $\ell_2$ | FAB | 10.22 | 10.13 | 13.45 | 36.72 | - | - | - | - | 84.27 | 109.43 | 108.87 | - | - | - |
| | CW | 4.22 | 4.09 | 5.17 | 10.07 | 4.23 | 4.14 | 5.15 | 10.06 | 25.90 | 31.32 | 31.31 | 25.78 | 31.32 | 31.30 |
| | BB | 4.44 | 4.15 | 5.03 | 12.38 | 26.20 | 26.76 | 27.24 | 31.00 | 26.64 | 31.82 | 31.90 | 48.74 | 54.35 | 54.07 |
| | DDN | **3.42** | **3.33** | **4.30** | **8.59** | **3.42** | **3.35** | **4.32** | **8.60** | **24.14** | **29.62** | **29.48** | **23.61** | **29.61** | **29.52** |
| | Ours | 4.46 | 4.42 | 5.48 | 9.15 | 4.50 | 4.44 | 5.47 | 9.09 | 24.88 | 30.22 | 30.08 | 25.39 | 30.21 | 30.04 |
| $\ell_\infty$ | FAB | 10.85 | 10.61 | 14.05 | 36.23 | - | - | - | - | 84.62 | 109.83 | 109.57 | - | - | - |
| | BB | 14.26 | 16.36 | 13.51 | 15.44 | 38.61 | 38.87 | 36.39 | 34.85 | 61.34 | 62.36 | 62.63 | 83.70 | 87.64 | 88.90 |
| | Ours | **4.25** | **4.33** | **5.30** | **9.17** | **4.33** | **4.23** | **5.31** | **9.10** | **24.84** | **30.15** | **30.01** | **24.78** | **30.19** | **30.03** |

## 3.2. Experimental Results

**Perturbation size.** Table 1 reports the median value of $\|\delta^\star\|$ at $Q = 1000$ queries, for all models, attacks and norms. The values obtained confirm that our attack can find smaller or comparable perturbations with those found by the competing attacks, in most of the untargeted and targeted cases, and that the biggest margin is achieved in the $\ell_1$ case. FMN is only slightly worse than DDN and BB in a few cases, including $\ell_2$-DDN on M4 and $\ell_\infty$-BB on M2. The reason may be that these robust models exhibit noisy gradients and flat regions around the clean input samples, hindering the initial optimization steps of the FMN attack.

**Execution time.** The average runtime per query for each attack-model pair, measured on a workstation with an NVIDIA GeForce RTX 2080 Ti GPU with 11GB of RAM, can be found in Table 2. Our attack is up to 2-3 times faster, with the exception of DDN in the $\ell_2$ case. This is compensated by the fact that FMN finds better solutions.

## 4. Conclusions

This work introduces a novel minimum-norm attack that combines all desirable traits to help improve current adversarial evaluations: (i) finding smaller or comparable minimum-norm perturbations across a range of models and datasets; (ii) reducing runtime up to 3 times per query with respect to competing attacks. FMN also works with different $\ell_p$ norms ($p = 0, 1, 2, \infty$) and it does not necessarily require initialization from the target class.

We firmly believe that FMN will establish itself as a useful tool in the arsenal of robustness evaluation. By facilitating more reliable robustness evaluations, we expect that FMN will foster advancements in the development of machine-learning models with improved robustness guarantees.

## Acknowledgements

This work has been partly supported by the PRIN 2017 project RexLearn (grant no. 2017TWNMH2), funded by the Italian Ministry of Education, University and Research; and by BMK, BMDW, and the Province of Upper Austria in the frame of the COMET Programme managed by FFG in the COMET Module S3AI. WB acknowledges support from the German Federal Ministry of Education and Research (BMBF) through the Competence Center for Machine Learning (TUE.AI, FKZ 01IS18039A), from the German Science Foundation (DFG) under grant no. BR 6382/1-1 (Emmy Noether Program) as well as support by Open Philantropy and the Good Ventures Foundation.

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

# Appendix

## A. Hyperparameters

We select the hyperparameters to be optimized for each attack as recommended by the corresponding authors (Brendel et al., 2019; Carlini & Wagner, 2017; Croce & Hein, 2020b; Rony et al., 2019). For attacks that are claimed to be robust to hyperparameter changes, like BB and FAB, we follow the recommendation of using a larger number of random restarts rather than increasing the number of hyperparameter configurations to be tested. In addition, as BB requires being initialized from an adversarial starting point, we initialize it by randomly selecting a sample either from a different class (in the untargeted case) or from the target class (in the targeted case). Finally, as each attack performs operations with different levels of complexity within each iteration, possibly querying the model multiple times, we set the number of steps for each attack such that at least $1,000$ forward passes (i.e., *queries*) are performed. This ensures a fairer comparison also in terms of the computational time and resources required to execute each attack.

*CW.* This attack minimizes the soft-constraint version of our problem, i.e., $\min_{\boldsymbol{\delta}} \|\boldsymbol{\delta}\|_p + c \cdot \min(L(\boldsymbol{x} + \boldsymbol{\delta}, y, \boldsymbol{\theta}), -\kappa)$. The hyperparameters $\kappa$ and $c$ are used to tune the trade-off between perturbation size and misclassification confidence. To find minimum-norm perturbations, CW requires setting $\kappa = 0$, while the constant $c$ is tuned via binary search (re-running the attack at each iteration). We set the number of binary-search steps to 9, and the maximum number of iterations to 250, to ensure that at least $1,000$ queries are performed. We also set different values for $c, \eta \in \{10^{-3}, 10^{-2}, 10^{-1}, 1\}$.

*DDN.* This attack, similarly to ours, maximizes the misclassification confidence within an $\epsilon$-sized constraint, while adjusting $\epsilon$ to minimize the perturbation size. We consider initial values of $\epsilon_0 \in \{0.03, 0.1, 0.3, 1, 3\}$, and run the attack with a different number of iterations $K \in \{200, 1000\}$, as this affects the size of each update on $\boldsymbol{\delta}$.

*BB.* This attack starts from a randomly-drawn adversarial point, performs a 10-step binary search to find a point which is closer to the decision boundary, and then updates the point to minimize its perturbation size by following the decision boundary. In each iteration, BB computes the optimal update within a given trust region of radius $\rho$. We consider different values for $\rho \in \{10^{-3}, 10^{-2}, 10^{-1}, 1\}$, while we fix the number of steps to 1000. We run the attack 3 times by considering different initialization points, and eventually retain the best solution.

*FAB.* This attack iteratively optimizes the attack point by linearly approximating its distance to the decision boundary. It uses an adaptive step size bounded by $\alpha_{\max}$ and

an extrapolation step $\eta$ to facilitate finding adversarial points. As suggested by Croce & Hein (2020b), we tune $\alpha_{\max} \in \{0.1, 0.05\}$ and $\eta \in \{1.05, 1, 3\}$. We consider 3 different random initialization points, and run the attack for 500 steps each time, eventually selecting the best solution.

*FMN.* We run FMN for $K = 1000$ steps, using $\gamma_0 \in \{0.05, 0.3\}$, $\gamma_K = 10^{-4}$, and $\alpha_K = 10^{-5}$. For $\ell_0$, $\ell_1$, and $\ell_2$, we set $\alpha_0 \in \{1, 5, 10\}$. For $\ell_\infty$, we set $\alpha_0 \in \{10^1, 10^2, 10^3\}$, as the normalized $\ell_2$ step yields much smaller updates in the $\ell_\infty$ norm. For each hyperparameter setting we run the attack twice, starting from (i) the input sample and (ii) an adversarial point.

## B. Additional experiments

**Evaluation criteria.** We evaluate the attacks along four different criteria: (i) *perturbation size* and (ii) *robustness to hyperparameter selection*, measured as the median $\|\boldsymbol{\delta}^\star\|_p$ on the test set (for a fixed budget of $Q$ queries and for sample- and dataset-level hyperparameter tuning); (iii) *execution time*, measured as the average time spent per query (in milliseconds); and (iv) *convergence speed*, measured as the average number of queries required to converge to a good-enough solution (within 10% of the best value found at $Q = 1000$).

**Sample-level vs. Dataset-level tuning.** In order to inspect the capabilities of our attack, we consider two main scenarios: tuning the hyperparameters at the *sample-level* and at the *dataset-level*. In the sample-level scenario, we select the optimal hyperparameters separately for each input sample. In the dataset-level scenario, we choose the same hyperparameters for all samples, selecting the configuration that yields the best attack performance. While sample-level tuning provides a fairer comparison across attacks, it is more computationally demanding and less practical than dataset-level tuning. In addition, the comparison allows us to understand how robust attacks are to suboptimal hyperparameter choices.

*Query-distortion (QD) curves.* To evaluate each attack in terms of perturbation size under the same query budget $Q$, we use the so-called QD curves introduced by Brendel et al. (2019). These curves report, for each attack, the median value of $\boldsymbol{\delta}^\star$ as a function of the number of queries $Q$. For each given $Q$ value, the optimal $\boldsymbol{\delta}^\star$ for each point is selected among the different attack executions (i.e., using different hyperparameters and/or initialization points, as described in Sect. 3.1). In Fig. 2, we report the QD curves for the MNIST and CIFAR10 challenge models (i.e., M2 and C1) in the untargeted scenario. It is worth noting that our attack attains comparable results in terms of perturbation size across all norms, while significantly outperforming FAB and BB in the $\ell_1$ case. It typically requires also less iterations

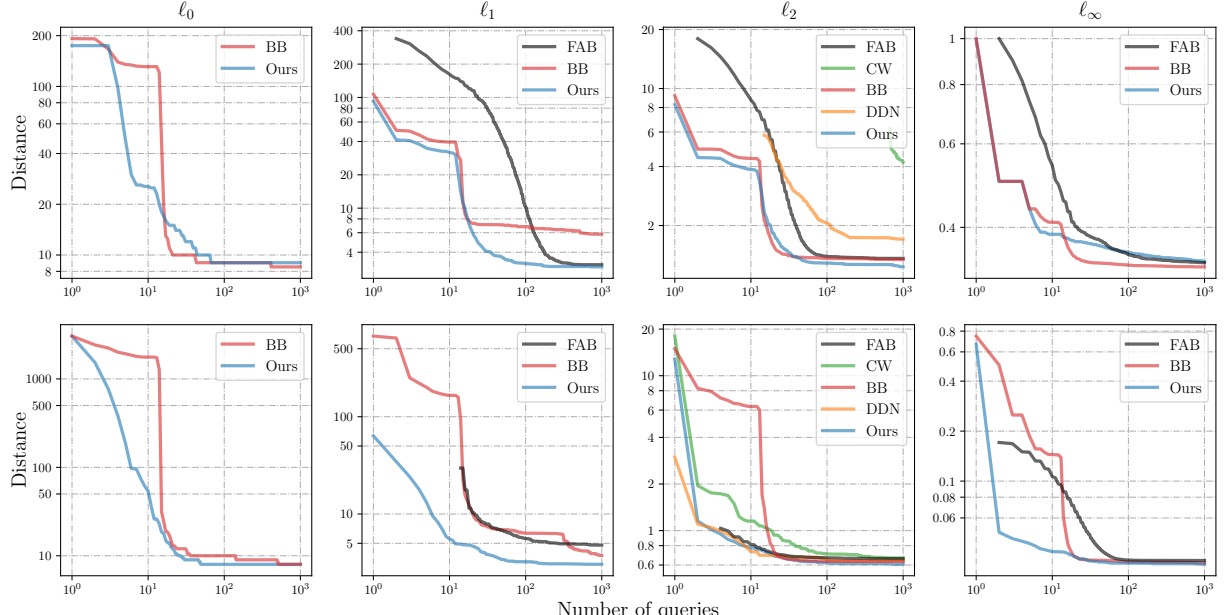

*Figure 2.* Query-distortion curves for MNIST (M2) and CIFAR10 (C1) models (untargeted scenario).

than the other attacks to converge. While the QD curves show the complete behavior of each attack as $Q$ increases, a more compact and thorough summary of our evaluation is reported in Table 3, according to the four evaluation criteria described. The remaining curves, computed for all models, can be found in the end of the Appendix.

**Robustness to hyperparameter selection.** The values reported in the lower part of Table 3 show that, when using dataset-level hyperparameter tuning, FMN outperforms the other attacks in a much larger number of cases. This shows that FMN is more robust to hyperparameter changes, while other attacks like $\ell_0$- and $\ell_1$-BB suffer when using the same hyperparameters for all samples.

**Convergence speed.** To get an estimate of the convergence speed, we measure the number of queries required by each attack to reach a perturbation size that is within 10% of the value found at $Q = 1000$ queries (the lower the better). Results are shown in Table 4. Our attack converges on par with or faster than all other attacks for almost all models, often requiring only half or a fifth as many queries as the state of the art.

**Experiments on ImageNet.** We expand further our experiments by running an additional comparison between FMN and a widely-used maximum-confidence attack, i.e., the Projected Gradient Descent (PGD) attack (Madry et al., 2017), on two pretrained ImageNet models (i.e., ResNet18 and VGG16), considering $\ell_1$, $\ell_2$ and $\ell_\infty$ norms. The hyperparameters are tuned at the *dataset-level* using 20 validation samples. For FMN, we fix the hyperparameters as discussed

before, and only tune $\alpha_0 \in \{0.1, 1, 2, 8\}$, without using adversarial initialization. For PGD, we tune the step size $\alpha \in \{0.001, 0.01, 0.1, 1, 2, 8\}$. We run both attacks for $Q = 1,000$ queries on a separate set of $1,000$ samples. The success rates of both attacks at fixed $\epsilon$ values are reported in Table 5. The results show that FMN outperforms or equals PGD in all norms.

*Table 5.* Success rate (%) of FMN against PGD on ImageNet models.

|  |  | ResNet18 | VGG |
|---|---|---|---|
| $\ell_1$ ($\epsilon = 1.0$) | PGD | 31.4 | 30.4 |
|  | FMN | **38.4** | **39.8** |
| $\ell_2$ ($\epsilon = 0.15$) | PGD | 61.7 | 61.4 |
|  | FMN | **65.8** | **66.2** |
| $\ell_\infty$ ($\epsilon = 4 \cdot 10^{-4}$) | PGD | 51.0 | **49.0** |
|  | FMN | **55.2** | **49.0** |

## C. Adversarial Examples

In Figs. 3-4, we report adversarial examples generated by all attacks against model M2 and C2, respectively, on MNIST and CIFAR10 datasets, in the untargeted scenario.

The clean samples and the original label are displayed in the first row of each figure. In the remaining rows we show the perturbed sample along with the predicted class and the corresponding norm of perturbation $\|\boldsymbol{\delta}^\star\|_p$. It is worth noting that the output class for different untargeted attacks is not always the same, which might sometimes explain

*Table 3.* Median $\|\delta^\star\|_p$ value at $Q = 1000$ queries for targeted and untargeted attacks, with sample-level and dataset-level hyperparameter tuning.

| | | MNIST | | | | | | | | CIFAR10 | | | | | |
|---|---|---|---|---|---|---|---|---|---|---|---|---|---|---|---|
| | | *Untargeted* | | | | *Targeted* | | | | *Untargeted* | | | *Targeted* | | |
| | *Model* | M1 | M2 | M3 | M4 | M1 | M2 | M3 | M4 | C1 | C2 | C3 | C1 | C2 | C3 |
| | | *Sample-level Hyperparameter Tuning* | | | | | | | | | | | | | |
| $\ell_0$ | BB | **7** | **8** | **15** | 94 | **14** | 27 | **24** | 93 | **8** | 12 | **13** | **19** | **32** | **25** |
| | Ours | **7** | 9 | **15** | **5** | **14** | **20** | **24** | **23** | **8** | **11** | 14 | **19** | **32** | 27 |
| $\ell_1$ | FAB | 6.60 | 3.08 | 14.23 | 109.4 | - | - | - | - | 4.79 | 5.17 | 8.79 | - | - | - |
| | BB | 6.26 | 5.81 | 13.16 | 5.44 | 12.42 | 10.38 | 20.41 | **6.25** | 3.75 | 4.29 | 8.62 | 8.04 | 10.93 | 15.71 |
| | Ours | **5.57** | **2.95** | **12.04** | **1.96** | **12.20** | **6.75** | **18.79** | 7.31 | **3.04** | **3.43** | **8.26** | **7.07** | **9.40** | **15.24** |
| $\ell_2$ | FAB | 1.45 | 1.36 | 2.62 | 2.97 | - | - | - | - | 0.66 | 0.72 | 0.94 | - | - | - |
| | CW | 1.49 | 4.22 | 2.78 | - | 2.33 | 6.97 | 3.54 | - | 0.67 | 0.74 | **0.91** | 1.08 | 1.27 | 1.38 |
| | BB | 1.43 | 1.34 | 2.61 | 1.61 | **2.27** | 2.04 | 3.23 | 1.79 | 0.63 | 0.70 | **0.91** | 1.07 | 1.26 | 1.38 |
| | DDN | 1.46 | 1.71 | 2.56 | **0.79** | 2.29 | 2.20 | 3.27 | **1.33** | 0.64 | 0.73 | **0.91** | 1.09 | 1.29 | 1.39 |
| | Ours | **1.41** | **1.23** | **2.50** | 0.94 | 2.28 | **1.89** | 3.19 | 1.85 | **0.61** | **0.69** | **0.91** | **1.03** | **1.21** | 1.38 |
| $\ell_\infty$ | FAB | .138 | .337 | .233 | .421 | - | - | - | - | .033 | .043 | .025 | - | - | - |
| | BB | .138 | **.330** | .227 | **.402** | .202 | **.355** | **.271** | **.403** | .032 | .041 | **.024** | **.055** | .064 | **.037** |
| | Ours | **.134** | .339 | **.226** | .404 | **.201** | .389 | .272 | .406 | **.032** | **.040** | **.024** | **.055** | **.063** | **.037** |
| | | *Dataset-level Hyperparameter Tuning* | | | | | | | | | | | | | |
| $\ell_0$ | BB | 12 | 152 | 52 | 145 | 20 | 179 | 39 | 183 | 28 | 44 | 32 | 29 | 65 | 33 |
| | Ours | **9** | **33** | **18** | **15** | **16** | **48** | **28** | **55** | **11** | **17** | **16** | **25** | **38** | **32** |
| $\ell_1$ | FAB | 8.66 | 225.7 | 163.9 | 312.3 | - | - | - | - | - | - | 20.48 | - | - | - |
| | BB | 10.60 | 49.83 | 17.57 | 46.99 | 16.60 | 53.11 | 29.89 | 54.31 | 7.02 | 10.20 | 17.13 | 11.41 | 15.26 | 23.37 |
| | Ours | **7.13** | **4.18** | **13.66** | **4.99** | **13.18** | **8.33** | **21.37** | **12.16** | **4.28** | **4.82** | **9.52** | **8.51** | **10.40** | **17.32** |
| $\ell_2$ | FAB | 1.54 | 1.59 | 2.81 | 16.30 | - | - | - | - | 0.77 | 1.11 | 1.06 | - | - | - |
| | CW | 1.63 | 5.15 | 3.71 | - | 2.50 | - | 4.72 | - | 0.86 | 1.00 | 0.99 | 1.36 | 2.90 | 1.55 |
| | BB | 1.75 | 1.82 | 3.02 | 4.57 | 2.64 | 2.59 | 3.52 | 5.31 | 0.86 | 0.95 | 1.10 | 1.25 | 1.45 | 1.73 |
| | DDN | **1.47** | 2.01 | 2.62 | **1.15** | 2.31 | 2.72 | 3.36 | **1.96** | **0.66** | 0.77 | 0.91 | 1.11 | 1.31 | 1.40 |
| | Ours | 1.61 | **1.42** | **2.61** | 1.56 | **2.30** | **2.13** | **3.24** | 2.41 | 0.67 | **0.74** | **0.91** | **1.09** | **1.28** | **1.38** |
| $\ell_\infty$ | FAB | .148 | .365 | .248 | .900 | - | - | - | - | .038 | .052 | .029 | - | - | - |
| | BB | .159 | **.336** | .243 | .409 | .223 | **.361** | .280 | .477 | .044 | .054 | .029 | .059 | .074 | .042 |
| | Ours | **.140** | .357 | **.233** | **.408** | **.206** | .426 | **.277** | **.434** | **.034** | **.042** | **.024** | **.057** | **.066** | **.037** |

*Table 4.* Number of queries required by each attack to reach a perturbation size that is within 10% of the value obtained at $Q = 1000$.

| | | MNIST | | | | | | | | CIFAR10 | | | | | |
|---|---|---|---|---|---|---|---|---|---|---|---|---|---|---|---|
| | | *Untargeted* | | | | *Targeted* | | | | *Untargeted* | | | *Targeted* | | |
| | *Model* | M1 | M2 | M3 | M4 | M1 | M2 | M3 | M4 | C1 | C2 | C3 | C1 | C2 | C3 |
| $\ell_0$ | BB | **22** | **43** | 68 | **114** | 30 | 443 | 71 | 376 | 497 | 372 | 58 | 384 | 500 | 85 |
| | Ours | **22** | 82 | **38** | 182 | **27** | **165** | **46** | **145** | **48** | **71** | **37** | **271** | **146** | **70** |
| $\ell_1$ | FAB | 44 | **242** | 152 | 569 | - | - | - | - | 124 | 220 | 72 | - | - | - |
| | BB | 24 | 314 | 83 | **391** | 45 | 614 | 233 | 722 | 674 | 570 | 34 | 526 | 464 | 206 |
| | Ours | **21** | 363 | **34** | 631 | **25** | **243** | **37** | **336** | **48** | **85** | **31** | **89** | **130** | **38** |
| $\ell_2$ | FAB | 14 | 60 | 40 | 532 | - | - | - | - | 18 | 28 | 14 | - | - | - |
| | CW | 110 | 799 | 335 | - | 100 | 913 | 469 | - | 67 | 39 | 33 | 56 | 144 | 42 |
| | BB | 20 | **24** | 20 | 337 | 21 | **61** | 20 | 692 | 22 | 23 | 22 | 26 | 27 | 29 |
| | DDN | **12** | 136 | **15** | 474 | 12 | 149 | 26 | 670 | **13** | **20** | **4** | **18** | **19** | 18 |
| | Ours | 16 | 94 | 16 | **190** | **11** | 136 | **16** | **188** | 28 | 23 | 7 | 25 | 29 | **13** |
| $\ell_\infty$ | FAB | 36 | 50 | 44 | 11 | - | - | - | - | 50 | 50 | 54 | - | - | - |
| | BB | 19 | 17 | **20** | **5** | **24** | 17 | **22** | **5** | **20** | 24 | 21 | 27 | 33 | **29** |
| | Ours | **9** | **10** | 22 | **5** | 27 | **8** | 26 | **5** | 22 | **15** | **14** | **20** | **29** | 34 |

differences in the perturbation sizes. An example is given in Fig. 4b, where the sample in the fourth column, labeled as "ship", is perturbed by most of the attacks towards the class "airplane", while in our case it outputs the class "dog" with a much smaller distance.

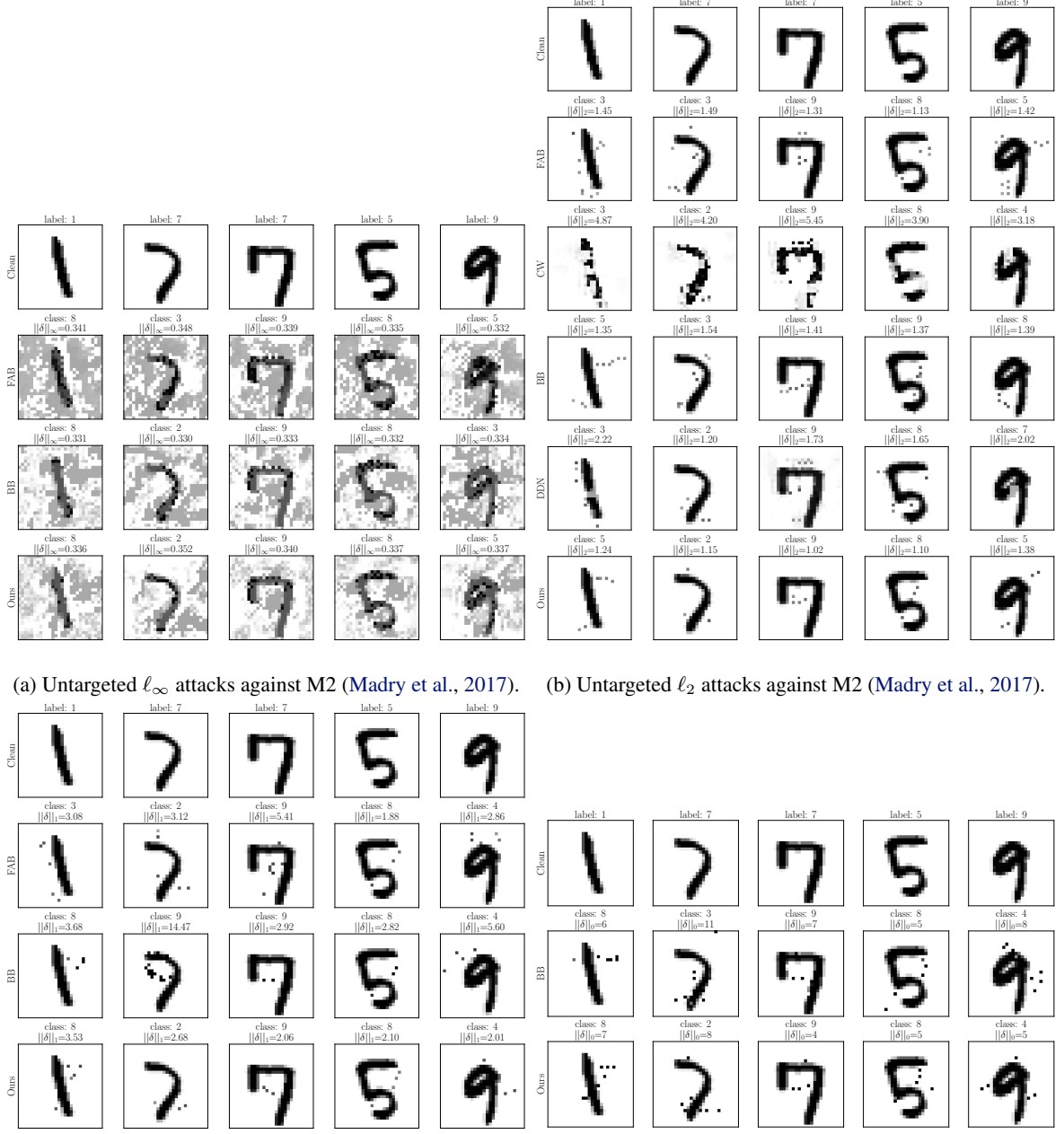

(a) Untargeted $\ell_\infty$ attacks against M2 (Madry et al., 2017).

(b) Untargeted $\ell_2$ attacks against M2 (Madry et al., 2017).

(c) Untargeted $\ell_1$ attacks against M2 (Madry et al., 2017).

(d) Untargeted $\ell_0$ attacks against M2 (Madry et al., 2017).

*Figure 3.* Adversarial examples on MNIST dataset.

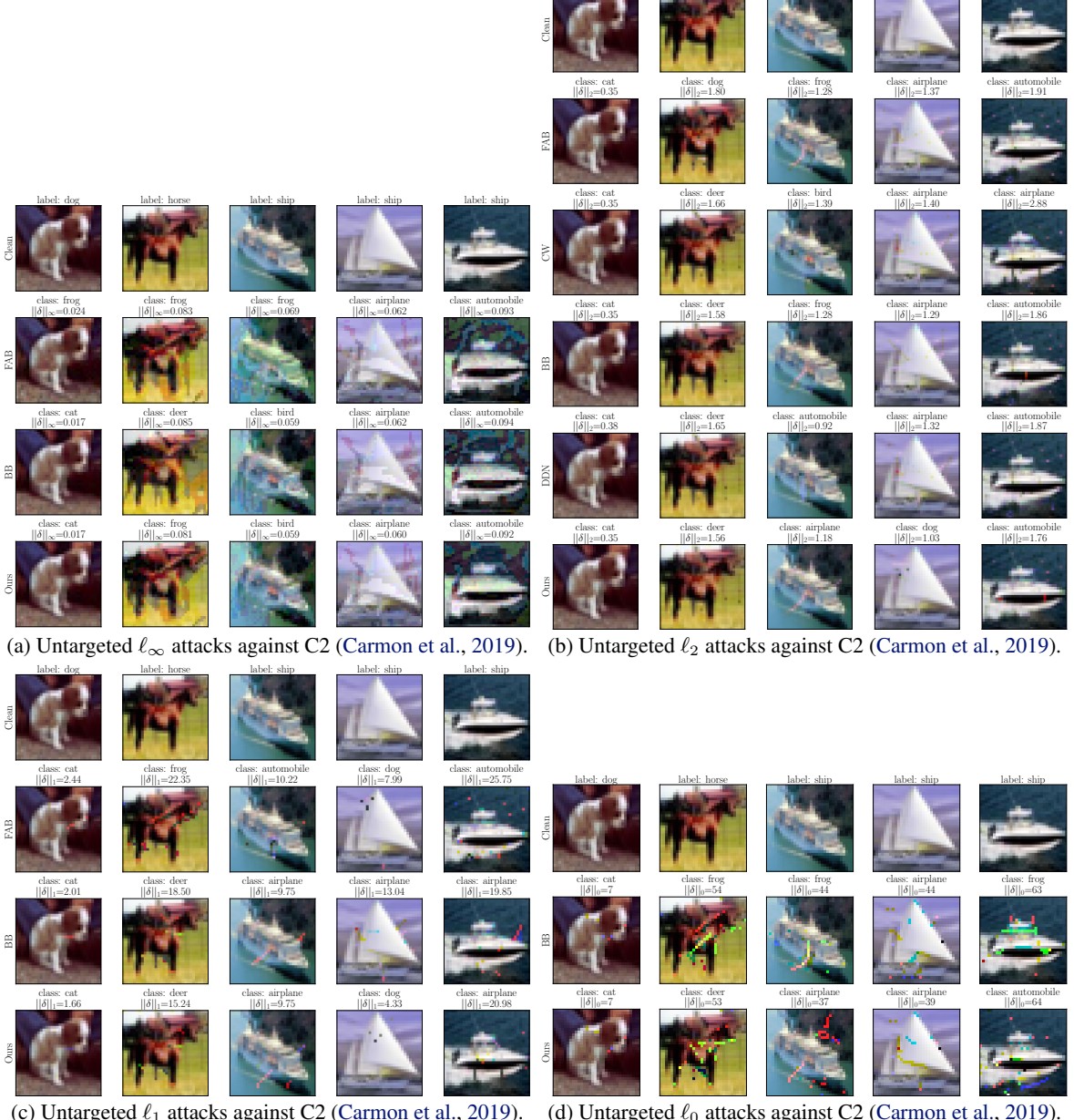

(a) Untargeted $\ell_\infty$ attacks against C2 (Carmon et al., 2019). (b) Untargeted $\ell_2$ attacks against C2 (Carmon et al., 2019).

(c) Untargeted $\ell_1$ attacks against C2 (Carmon et al., 2019). (d) Untargeted $\ell_0$ attacks against C2 (Carmon et al., 2019).

*Figure 4.* Adversarial examples on CIFAR10 dataset.

# D. Additional query-distortion Curves

We already introduced in sect. B the query-distortion curves as an efficiency evaluation metric for the attacks. We report here the complete results for all models, in targeted and untargeted scenarios.

On the MNIST dataset, our attacks generally reach smaller norms with fewer queries, with the exception of M2 (Figs. 5-6), where it seems to reach convergence more slowly than BB in $\ell_0$ and $\ell_\infty$. In $\ell_2$, the CW attack is the slowest to converge, due to the need of carefully tuning the weighting term, as described in Sect. 1.

On the CIFAR10 dataset (Figs. 7-8), our attack always rivals or outperforms the others, with the notable exception of DDN for the $\ell_2$ norm, which sometimes finds smaller perturbations more quickly, as also shown in Table 4.

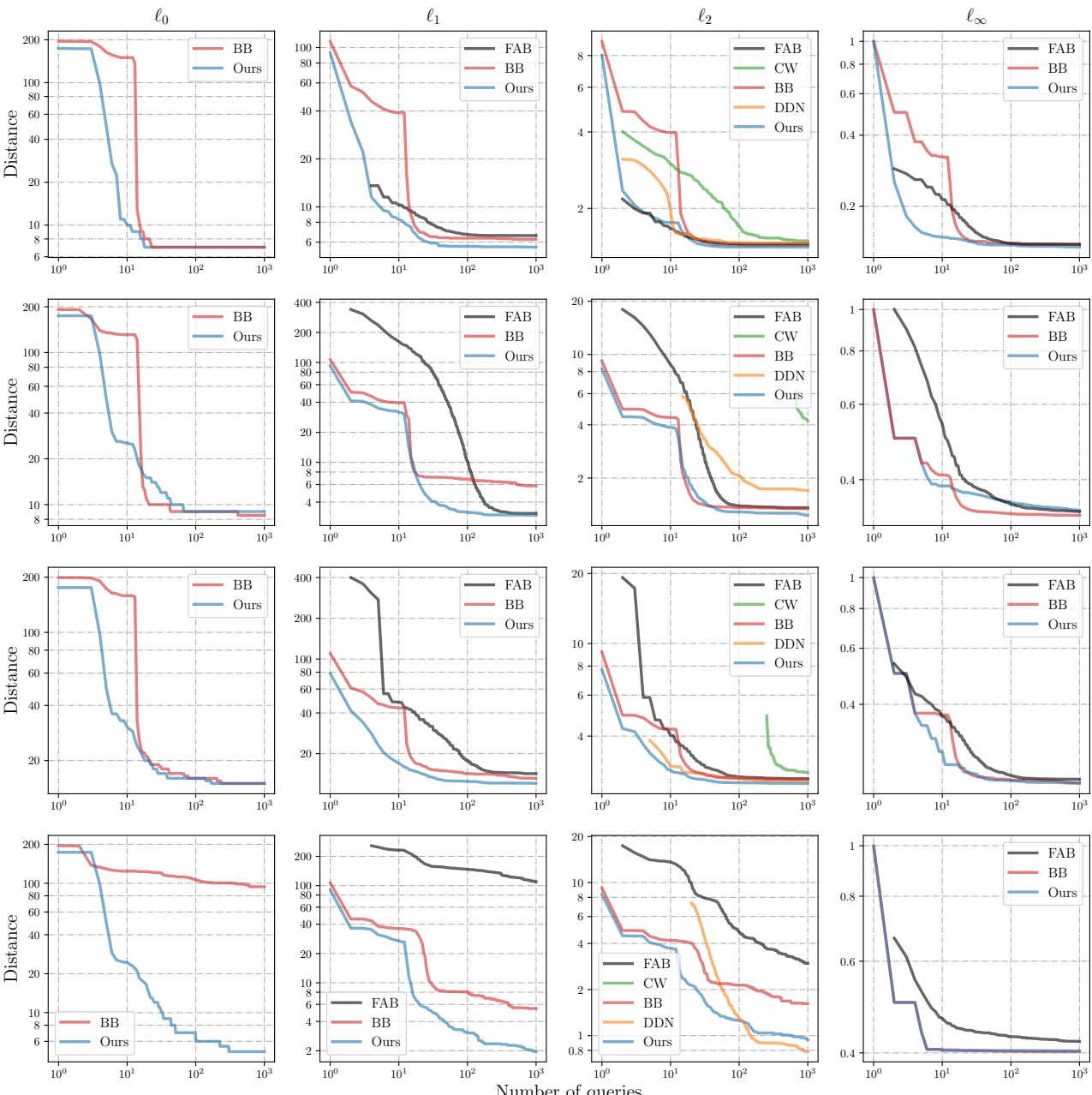

Figure 5. Query-distortion curves for untargeted (U) attacks on the M1, M2, M3, and M4 MNIST models.

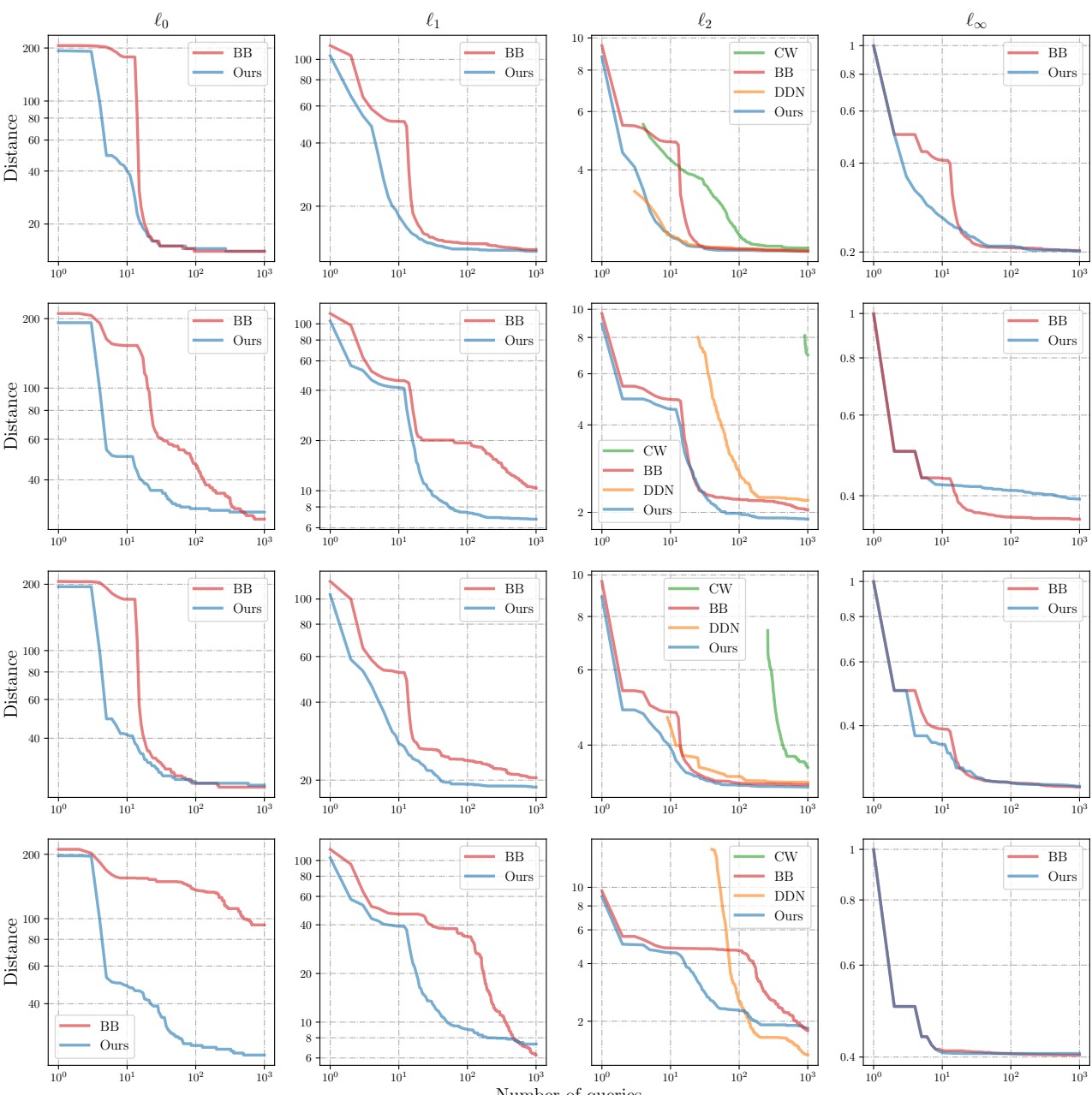

*Figure 6.* Query-distortion curves for targeted (*T*) attacks on the M1, M2, M3 and M4 MNIST models.

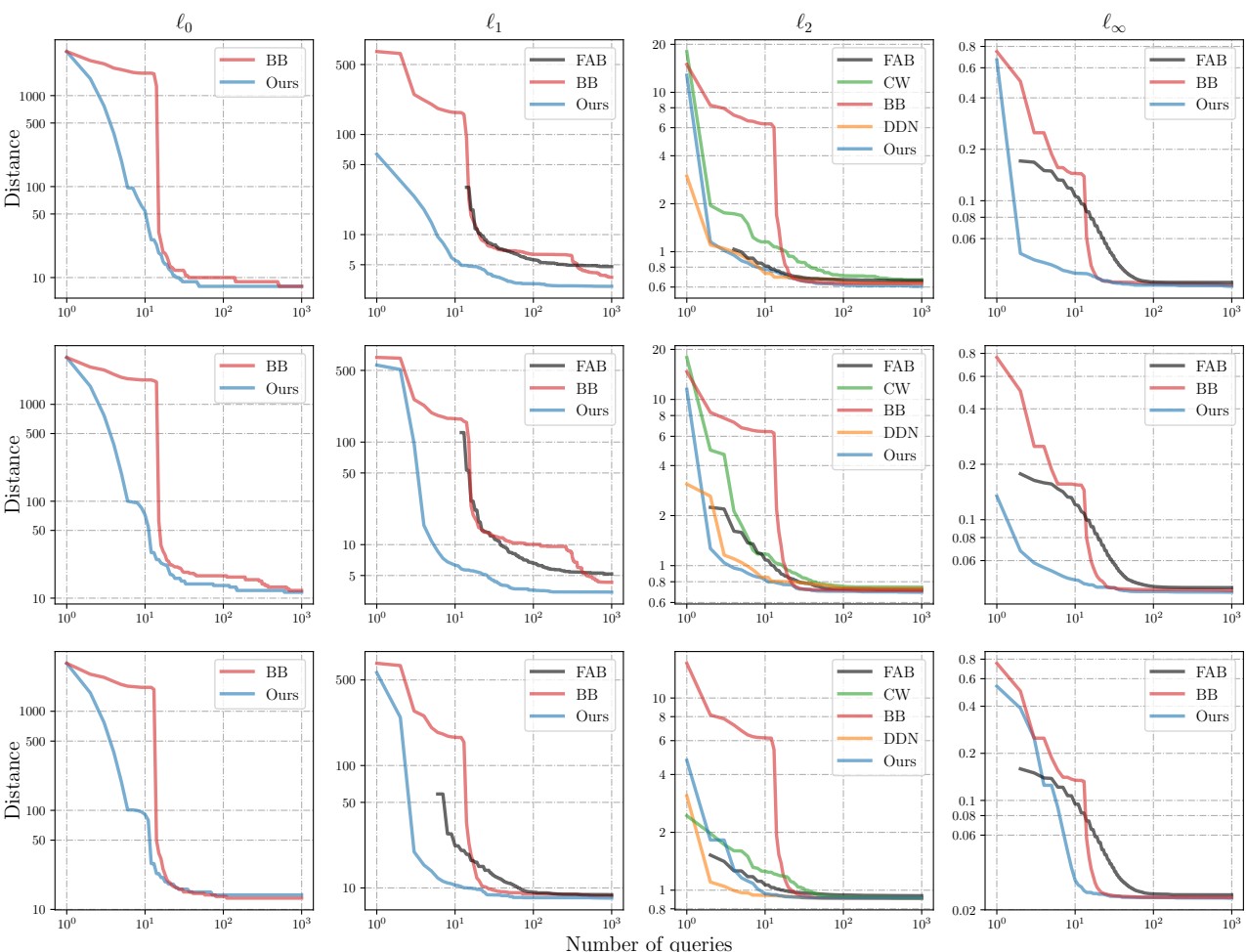

*Figure 7.* Query-distortion curves for untargeted (*U*) attacks on the C1 (*top*), C2 (*middle*), and C3 (*bottom*) CIFAR10 models.

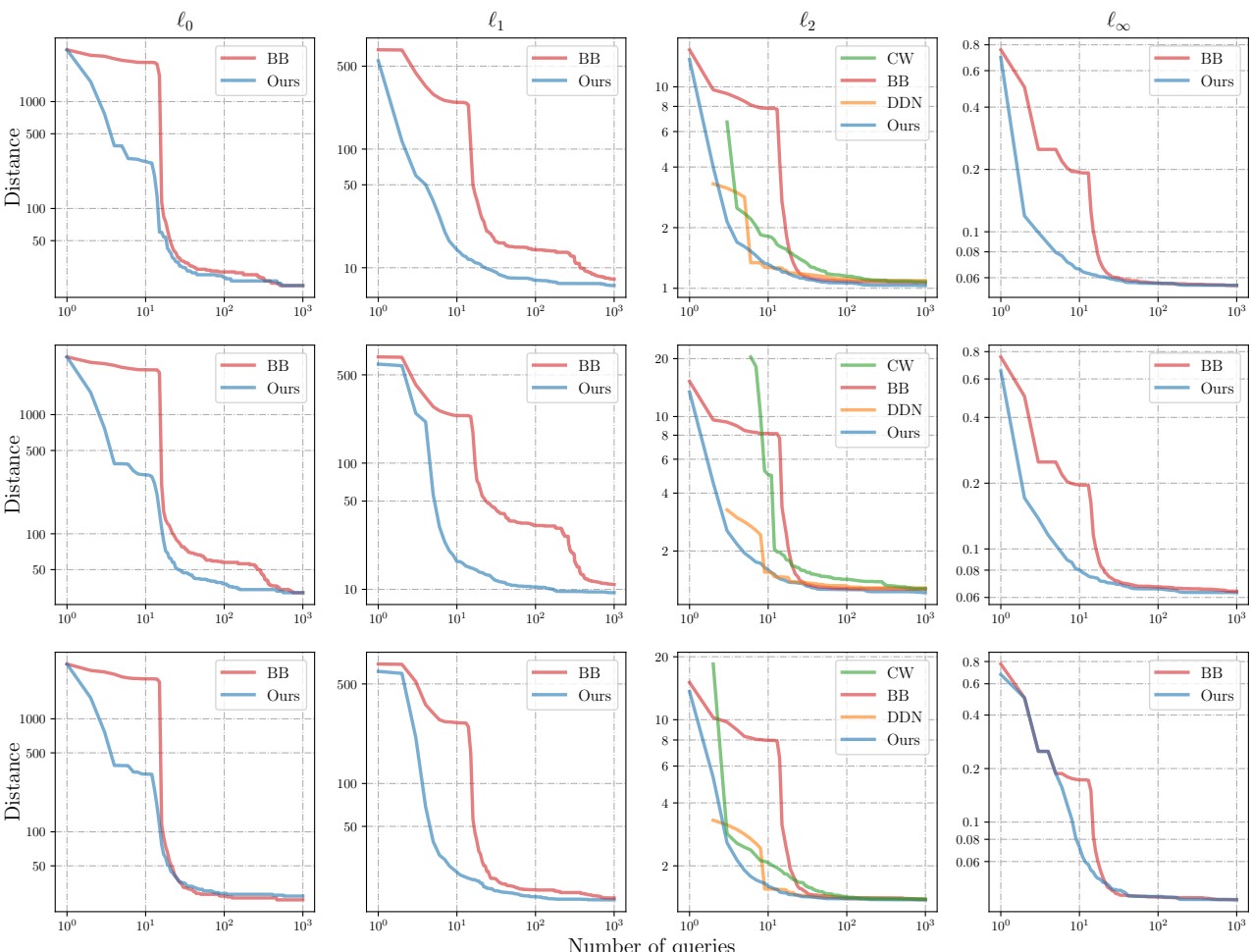

*Figure 8.* Query-distortion curves for targeted (*T*) attacks on the C1 (*top*), C2 (*middle*), and C3 (*bottom*) CIFAR10 models.