# OpenReview forum: "Fast Minimum-norm Adversarial Attacks through Adaptive Norm Constraints"
_ICML.cc/2021/Workshop/AML — ICML 2021 Workshop AML Oral_

### Official Review · Reviewer_ptSG · 2021-06-20
**A new good performing white-box attack method for different distance metrics**

**Rating:** Accept
**Confidence:** 3

**Review:**

The authors propose a good performing FMN attack, which keeps the advantages of the DDN, while supports $\ell_p$ norms other than $\ell_2$. This attack achieves fast convergence speed and small perturbation sizes compared with state-of-the-art attacks. It should be a useful tool when carrying out robustness evaluations.

This paper is clear.

Pros:
1. The evaluation experiments are designed carefully and comprehensively.
2. The proposed attack achieves comparable or even better performance than state-of-the-art methods in terms of computational time, convergence speed, and perturbation size.
3. The proposed attack works well under different $\ell_p$-norm consistently, especially for the $\ell_1$-norm.

Cons:
1. Though this work has differences from the DDN and fixed several flaws in DDN, its intuition of decoupling direction and norm seems to be quite similar to DDN.
2. Many existing works usually use the average perturbation size as the evaluation criteria instead of its median. Including the average perturbation size as the evaluation criteria would allow comparing to existing works more easily.

---

### Decision · Program_Chairs · 2021-06-21

**Decision:**

Accept (Oral)

**Comment:**

This paper proposed a good performing FMN attack, achieving fast convergence speed and small perturbation sizes compared with state-of-the-art attacks. There are many pros as agreed by the reviewer.